# Co-Supplementation of Diet with *Saccharomyces cerevisiae* and Thymol: Effects on Growth Performance, Antioxidant and Immunological Responses of Rainbow Trout, *Oncorhynchus mykiss*

**DOI:** 10.3390/ani15030302

**Published:** 2025-01-22

**Authors:** Morteza Yousefi, Hossein Adineh, Ali Taheri Mirghaed, Seyyed Morteza Hoseini

**Affiliations:** 1Department of Veterinary Medicine, RUDN University, 6 Miklukho-Maklaya St., 117198 Moscow, Russia; myousefi81@gmail.com; 2Department of Fisheries, Faculty of Agriculture and Natural Resources, Gonbad Kavous University, Gonbad Kavous 4971799151, Iran; 3Department of Aquatic Animal Health, Faculty of Veterinary Medicine, University of Tehran, Tehran 1419963114, Iran; mirghaed@ut.ac.ir; 4Inland Waters Aquatics Resources Research Center, Iranian Fisheries Sciences Research Institute, Agricultural Research, Education and Extension Organization, Gorgan 4916687631, Iran

**Keywords:** thymol, *Saccharomyces cerevisiae*, gut transcription, humoral immunity, mucosal immunity, antioxidant

## Abstract

This study assessed the combined effects of dietary yeast and thymol on the growth and health of rainbow trout. The results showed that dietary yeast supplementation improves fish growth, antioxidant and immunological parameters. Dietary thymol supplementation does not improve fish growth but does improve antioxidant and immunological parameters. Dietary yeast and thymol supplementation shows interaction effects on the gut-immune-related transcripts. Based on the results, yeast supplementation was particularly effective in enhancing fish growth performance and non-specific immunity, while thymol supplementation primarily led to improved antioxidant capacity.

## 1. Introduction

Aquaculture is a rapidly expanding sector that plays a crucial role in the global food supply, with the rainbow trout, *Oncorhynchus mykiss*, emerging as a key candidate species. This is a carnivorous species, which is well-adapted to artificial diets and is reared in clean waters with temperature below 21 °C [1]. The global annual production rates of this species in inland and marine waters were 739,500 and 220,100 tones, respectively [2]. However, one significant challenge hindering the growth of aquaculture is the occurrence of disease outbreaks in fish farms, primarily attributed to the stressful conditions inherent in fish farming [3]. These diseases lead to economic losses, whether or not they can be effectively treated. Therefore, enhancing fish disease resistance is a logical strategy to prevent such outbreaks [4].

Thymol, a phenolic compound predominantly found in thyme essential oil, is recognized for its powerful antioxidant properties [5]. This compound, whether derived naturally from thyme or synthesized to be chemically identical, has been incorporated into aquafeeds and has demonstrated various benefits. For instance, supplementation with dietary thymol at concentrations ranging from 300 to 750 mg/kg has been shown to enhance growth, immunological responses and antioxidant parameters while reducing post-infection mortality in snakehead, *Channa argus* [6]. Additionally, thymol has been effective in alleviating the adverse effects of zinc toxicity in Nile tilapia, *Oreochromis niloticus*, supporting growth and improving both antioxidant and immunological parameters [7]. On the other hand, the research on the benefits of dietary thymol for rainbow trout is limited. One study indicated that thymol supplementation could improve growth rates, antioxidant and immunological parameters and disease resistance in juvenile rainbow trout (initial weight of 35 g) at high doses of up to 2500 mg/kg [8]. This underscores the need for further investigation into the potential applications of thymol in enhancing the health and resilience of rainbow trout within aquaculture systems.

*Saccharomyces cerevisiae*, a yeast extensively researched in fish nutrition, can be utilized in both its intact cell form and as extracted materials. This yeast is rich in beneficial compounds such as beta-glucan, mannan and nucleotides, which have been shown to enhance fish production, welfare and disease resistance [9]. When incorporated as a probiotic in fish diets, *S. cerevisiae* has demonstrated improvements in growth performance and overall health across various fish species [10,11]. Different strains of *S. cerevisiae* have been previously evaluated for their probiotic potential in rainbow trout. For instance, strain NCYC Sc 47/g was found to be ineffective in promoting growth rates in trout fingerlings; however, it did cause enhanced humoral nonspecific immunity [12]. This particular strain also failed to significantly modulate the gut microbiome in trout fry [13]. Conversely, other unidentified strains have shown promise in modulating the gut microbiome [14] and improving growth rates and overall health in juvenile trout [15].

While the individual effects of dietary thymol and *S. cerevisiae* have been explored in fish, their combined and potentially synergistic effects remain unstudied. The present study aims to investigate a novel strain of *S. cerevisiae* (PTCC 5052) as a probiotic for the first time in fish. Additionally, this research will assess the synergistic effects of dietary yeast and thymol on growth performance, antioxidant capacity and immunological responses and hindgut gene expression levels in fingerling rainbow trout.

## 2. Materials and Methods

### 2.1. Source of Thymol and Yeast

The thymol (98.5% purity) was obtained from Sigma-Aldrich Co. (St. Louis, MI, USA), while the lyophilized *S. cerevisiae* (PTCC 5052) was sourced from the Iranian Research Organization for Science and Technology (IROST, Tehran, Iran). The yeast was initially cultured in potato dextrose broth (PDB; Q-Lab Co., Westlake, Cleveland, OH, USA) at 25 °C for 24 h. Subsequently, inoculums from this culture were transferred to a new 1000 mL flask containing 700 mL of PDB medium. The flask was continuously shaken for an additional 24 h until the optical density of the suspension reached approximately 1.800 at 600 nm. At this stage, yeast cells were harvested via centrifugation (5000× *g* for 5 min) and re-suspended in sterile NaCl solution (0.85%). To assess the cell density in the suspension, serially diluted samples were cultured on potato dextrose agar (PDA; Q-Lab Co., Westlake, Cleveland, OH, USA) at 25 °C and the colony-forming units (cfu) were counted.

### 2.2. Diets

Thymol and yeast were incorporated into the fish diets at concentrations of 250 mg/kg and 500 mg/kg for thymol and 1 × 10^8^ cfu/g for yeast, arranged in a 3 × 2 factorial design. Hence, the diets were named CTL (control, no thymol or yeast), Y (1 × 10^8^ cfu/g yeast), TH250 (250 mg/kg thymol), TH250+Y (250 mg/kg thymol and 1 × 10^8^ cfu/g yeast), TH500 (500 mg/kg thymol) and TH500+Y (500 mg/kg thymol and 1 × 10^8^ cfu/g yeast). To prepare the diets, the feed ingredients (as detailed in Table 1) were thoroughly mixed for 20 min. A paste was created by adding 300 mL of water per kg of feed mixture. Feed pellets were then formed using a meat grinder and allowed to dry at room temperature. The yeast suspension was directly added to the water used for the paste, while the thymol was first dissolved in 500 µL of ethanol and mixed with sunflower oil before being incorporated into the feed mixture.

The yeast’s viability was evaluated every 20 days by preparing a suspension of each diet in a NaCl solution (0.85%). One gram of each diet was powdered and mixed with 10 mL of NaCl solution, then shaken for 20 min. The yeast count was determined using the micro-dilution method on PDA medium at 25 °C. The results indicated yeast counts of 2.14 ± 0.20 cfu_log_/g for CTL, 7.14 ± 0.17 cfu_log_/g for Y, 2.10 ± 0.07 cfu_log_/g for TH250, 7.10 ± 0.20 cfu_log_/g for TH250+Y, 1.86 ± 0.11 cfu_log_/g for TH500 and 7.07 ± 0.11 cfu_log_/g for TH500+Y.

### 2.3. Fish Husbandry and Growth Performance

The rainbow trout fingerlings (300 individuals, approximately 5 g each) were sourced from a private farm and acclimatized to the experimental conditions in a 1 m^3^ tank, where they were fed a commercial diet from Faradaneh Co. (Shahrekord, Iran). The acclimation period lasted 7 days with water temperature, dissolved oxygen, pH and total ammonia values of 13.2 ± 0.79 °C, 9.21 ± 0.89 mg/L, 7.21 ± 0.12 and 0.32 ± 0.09 mg/L, respectively. After acclimation, 270 visually uniform individuals (6.62 ± 0.18 g), free from morphological abnormalities and lesions, were randomly selected and distributed among 18 aquaria. Fifteen fish were stocked into each aquarium and three aquaria were assigned to each treatment (i.e., each aforementioned diet). Each aquarium contained 50 L of water and was equipped with aeration provided by an air stone, a biological filtration system consisting of 10 L of pumice (average diameter = 2 cm), a felt cloth as a physical filter and a 7 W UV lamp (Sobo Inc., Guangdong, China). Water circulation through the physical and biological filters was maintained using a water pump, with daily cleaning of the physical filters to remove fish waste. Additionally, 30% of the water in each aquarium was replaced daily with clean water. The water temperature, dissolved oxygen, pH and total ammonia values were 13.7 ± 0.33 °C, 8.05 ± 0.28 mg/L, 7.76 ± 0.08 and 0.42 ± 0.12 mg/L, respectively. A digital probe (Hach HQ40, Loveland, CO, USA) and a customized photometer (Palintest 7100, Gateshead, UK) were used for recording the water quality parameters.

Each diet was administered to three aquaria at a rate of 3% of biomass per day, divided into two meals. The feed amounts were adjusted biweekly based on the recorded aquarium biomass through bulk weighing. The feeding trial lasted for 60 days, after which the fish growth performance and feed efficiency were calculated as follows:
$$\mathrm{Specific}\;\mathrm{growth}\;\mathrm{rate}\;(\mathrm{SGR};\;\mathrm{as}\;\%/d)=100\;\times \;\frac{Ln\;(final\;weight)-Ln\;(initial\;weight)}{time\;(d)}$$

$$\mathrm{Feed}\;\mathrm{conversion}\;\mathrm{ratio}\;(\mathrm{FCR})=\frac{Consumed\;feed\;(g)}{Gained\;biomass\;(g)}$$

$$\mathrm{Weight}\;\mathrm{gain}\;(\mathrm{WG};\;\mathrm{as}\;\%)=100\;\times \;\frac{Final\;weight\;(g)-Initial\;weight\;(g)}{Initial\;weight\;(g)}$$


### 2.4. Sampling and Processing

At the conclusion of the feeding trial, three fish were randomly selected from each aquarium, anesthetized in a eugenol bath (50 ppm) and subjected to blood sampling via caudal puncture. The blood samples were collected using heparinized syringes and transferred into 2 mL plastic tubes. Following centrifugation at 5000× *g* for 10 min at 4 °C, the plasma samples were obtained and stored at −20 °C until further analysis.

Subsequently, the fish were euthanized with a sharp blow to the head followed by severing the spinal cord. The abdominal cavity was opened to extract the liver, which was immediately frozen in liquid nitrogen. The liver samples were then homogenized in a cold phosphate buffer (pH 7.0) at a ratio of 2 volumes (*w:v*) and centrifuged at 13,000 × *g* for 20 min at 4 °C. The resulting supernatants (enzyme extracts) were collected and stored at −70 °C until the analysis.

Skin mucus samples were also collected from three additional fish per aquarium. These fish were anesthetized in a eugenol bath and placed in a stainless steel tray. The skin mucus was harvested by rubbing a glass slide from the head to the tail of each fish. The collected mucus was transferred into a tube using another glass slide and rinsed with cold phosphate buffer (pH 7.0). After centrifugation at 13,000× *g* for 15 min at 4 °C, the supernatants were gathered and stored at −70 °C for the subsequent analysis.

Following mucus collection, the fish were euthanized as previously described and the abdominal cavity was opened to extract a segment of the hindgut, which was immediately frozen in liquid nitrogen for RNA extraction and gene expression assays.

### 2.5. Immunological Analysis

The plasma and mucus lysozyme activity was determined as described previously [16]. Briefly, *Micrococcus luteus* was suspended in a phosphate buffer (pH 6.2). The plasma/mucus samples (50 µL) were mixed with 1 mL of the bacterial suspension. The mixture was placed in a spectrophotometer and its optical density was recorded at 530 nm after 0.5 and 4.5 min. Each 0.001 decrease in optical density per min was equal to one unit of lysozyme activity [16].

The plasma and mucus total immunoglobulin (Ig) levels were determined according to Siwicki and Anderson [17] through the precipitation by 12% polyethylene glycol. The concentrations of soluble protein were determined in the samples before and after precipitation and subtracted to calculate the total Ig of the samples. The soluble proteins of the plasma and mucus samples were determined based on the Biuret and pyrogallol red methods, respectively.

The plasma alternative complement (ACH50) activity was determined based on the hemolytic activity against sheep erythrocyte [18]. Briefly, the plasma samples were 20 times diluted in a barbital buffer containing EGTA, magnesium and gelatin (pH 7.0–7.4). The diluted samples (100 µL) were mixed with the sheep erythrocyte (suspended in the barbital buffer) and incubated at room temperature for 90 min. Then, the percentage of hemolysis (relative to 100% hemolysis) was determined using a spectrophotometer at 412 nm.

The skin mucus alkaline phosphatase (ALP) activity was determined using a commercial kit (Zist Chem Co., Tehran, Iran) and spectrophotometer at 420 nm.

### 2.6. Bactericidal Activity

The bactericidal activities of the plasma and mucus were determined through the micro-dilution method. The *Aeromonas hydrophila*, *Yersinia ruckeri* and *Streptococcus iniae* were purchased from IROST (Tehran, Iran). A suspension of each bacterium was prepared in NaCl solution (0.85%) to give an optical density range of 0.05–0.06 at 600 nm. Equal volumes of the bacterial suspensions and plasma/mucus samples were mixed and incubated at room temperature for 2 h. Then, the mixtures were 10-fold serially diluted and cultured on tryptic soy agar (TSA; Q-Lab, Westlake, Cleveland, Ohio) for *A. hydrophila* and *Y. ruckeri* and on TSA + 10% blood for *S. iniae*. The plates were incubated at 25 °C for 24–48 h and the numbers of cfu were counted. The bactericidal activity was expressed as percentage of growth relative to a control (without plasma/mucus).

### 2.7. Plasma Enzymes

The plasma alanine aminotransferase (ALT), aspartate aminotransferase (AST) and lactate dehydrogenase (LDH) activities were determined using commercial kits (Zist Chem Co., Tehran, Iran). The decrease in NADPH concentrations in the reaction media was monitored at 340 nm and used for calculation of the enzymes’ activity, according to the manufacturer.

### 2.8. Hepatic Antioxidant Activity

The concentrations of soluble protein in the enzyme extracts were determined using a commercial kit (pyrogallol red protein assay kit; Zist Cehm Co., Tehran, Iran). The superoxide dismutase (SOD) activity was determined through the pyrogallol auto-oxidation method. One unit of SOD was equal to an amount of the enzyme inhibiting pyrogallol autoxidation by 50% [19]. The catalase (CAT) activity was determined by measuring the amount of hydrogen peroxide decomposition by the samples after 1 min. One unit of CAT activity was equal to an amount of the enzyme decomposing one molecule of hydrogen peroxide [20]. The glutathione peroxidase (GPx) activity was determined by measuring the consumption of reduced glutathione (GSH), which was added in excess to the reaction medium, for the decomposition of peroxide. The Ellman′s reagent was used as the chromogenic agent [21]. The GSH levels were measured via reaction with the Ellman’s reagent [22]. The total antioxidant capacity (TAC) and malondialdehyde (MDA) level were determined using commercial kits (Zellbio Co., Deutschland, Germany).

### 2.9. Hindgut Gene Expression

The frozen gut samples were used for RNA extraction. The procedures of RNA extraction, RNA quality and cDNA synthesis are described in [23] and detailed in the Appendix A. Gene expression was analyzed through amplification by real time PCR using specific primers for tumor necrosis factor-alpha (*tnf-a*), interleukin-1 beta (*il-1b*), transforming growth factor-beta (*tgf-b*), beta-defensin (*b-def*) and heat shock protein-70 (*hsp-70*) according to Table 2. The reaction mixture for the real-time PCR contained 7.5 µL of Taq SYBR green (Ampliqon 2x SYBR Master Mix Green, Odense, Denmark), 0.5 µL of forward/reverse primers and 1 µL of cDNA. The thermal reaction consisted of 15 min at 95 °C and 40 cycles (15 s at 94 °C, 40 s at 59 °C and 20 s at 72 °C). Each sample was analyzed by three technical replications. The cycle threshold (Ct) of each reaction was determined and used for calculations of the relative gene expression according to Livak and Schmittgen [24]. Glyceraldehyde 3-phosphate dehydrogenase (*gadph*) was used as the housekeeping gene for normalization, as it showed low variation among the treatments (2.29%).

### 2.10. Statistical Analysis

The data showed a normal distribution and homogenized variance, based on the Shapiro–Wilk and Levene tests, respectively. The gene expression data were log_2_-transformed and expressed as the fold change relative to the CTL treatment. A two-way ANOVA was used for analyzing the main and interaction effects of the dietary thymol and yeast on the tested parameters. The main effects of dietary thymol and interaction effects of dietary thymol × yeast were delineated by a Duncan test at *p* < 0.05. The data are presented as the mean ± standard deviation of three replicates. The software SPSS v.22 (IBM, SPSS Inc., Armonk, NY, USA) was used for the statistical analysis.

## 3. Results

### 3.1. Growth Performance

There were no fish mortalities in the treatments. The thymol showed no significant effects on fish growth performance or feed efficiency, whereas the dietary yeast significantly increased the final weight, SGR and WG and decreased the FCR. There were no interaction effects of dietary thymol and yeast on the growth performance or feed efficiency (Table 3).

### 3.2. Plasma Biochemical Parameters

The thymol and yeast showed no main or interaction effects on the plasma ALT and LDH activities. The plasma AST activities were significantly affected by both the dietary thymol and yeast. There was no significant difference in plasma AST activity between the 250 and 500 mg/kg thymol groups but both showed significantly lower enzyme activity than the 0 mg/kg thymol group (Table 4).

### 3.3. Hepatic Antioxidant Parameters

The dietary thymol significantly increased the GSH levels and TAC, SOD, CAT and GPx activities but decreased the MDA levels in the fish liver. There were no significant differences in these parameters between the 250 and 500 mg/kg thymol levels, except for GPx, which was significantly higher in the 250 mg/kg group than the 500 mg/kg group. The dietary yeast supplementation significantly increased the GSH levels and CAT activities but decreased the MDA levels in the fish liver. There were no interaction effects of thymol and yeast on the antioxidant parameters of the fish liver (Table 5).

### 3.4. Plasma and Mucus Immunological Parameters

The thymol and yeast had significant main effects on the plasma lysozyme activity. The dietary yeast supplementation significantly increased the plasma lysozyme activity; furthermore, the 250 and 500 mg/kg thymol groups showed similar plasma lysozyme levels, significantly higher than the 0 mg/kg thymol group. The plasma ACH50 activities and total Ig levels were significantly elevated by the dietary yeast inclusion but not thymol. No interaction effects of dietary yeast and thymol were found on the plasma immunological parameters (Table 6).

The skin mucus lysozyme activities were significantly elevated by the dietary yeast inclusion but not thymol. The activity of ALP in the skin mucus significantly increased with dietary thymol or yeast supplementation. The thymol-treated fish had similar mucosal ALP activities, being significantly higher than the fish fed the non-supplemented diets. The thymol and yeast showed no significant effects on the total Ig levels in the skin mucus. No interaction effects of dietary yeast and thymol were found on the mucosal immunological parameters (Table 6).

The dietary yeast supplementation significantly increased the bactericidal activity of the plasma against *A. hydrophila*, *Y. ruckeri* and *S. iniae*. The bactericidal activities of the plasma in the 250 and 500 mg/kg thymol groups were similar and significantly higher than those of the 0 mg/kg thymol groups. The thymol exhibited no significant effects on the bactericidal activity of the skin mucus against *A. hydrophila*, *Y. ruckeri* and *S. iniae*. No interaction effects of dietary thymol and yeast were observed on the bactericidal activities of the plasma and skin mucus against these bacteria (Figure 1).

### 3.5. Hidgut Immunological Transcripts

The dietary thymol and yeast supplementation showed interaction effects on the hindgut gene expression levels of *tnf-a*, *tgf-b*, *b-def* and *hsp-70*. There were up-regulations in the expression of *tnfa-a* in the Y, TH250+Y and TH500+Y treatments compared to the CTL treatment. The expression of *tgf-b*, *b-def* and *hsp-70* significantly increased in all treatments compared to the CTL treatment. The expression of *tgf-b* significantly decreased in the TH250+Y and TH500 groups compared to the other treatments. The expression of *b-def* and *hsp-70* significantly increased in the TH250+Y group compared to the TH250 treatment. The dietary thymol and yeast significantly up-regulated the *il-1b* gene expression, without any interaction. The gut *il-1b* expression levels in the CTL and TH250 groups were similar and significantly lower than that of the TH500 treatment (Figure 2).

## 4. Discussion

The study indicated that thymol does not function as a growth promoter for trout, which is in contrast to a previous study on this species [8]. Although the exact reasons for such differences between the studies are not clear, the fish starting weights may have contributed (6.5 vs. 35 g). Also, the previous study used very high concentrations of thymol (1000 to 2500 mg/kg), which were far higher than the optimum thymol requirement for other fish species [6,19]. However, these results present a need to assess the dietary thymol concentrations in trout with different individual weights. On the other hand, the current findings align with studies on Nile tilapia, which demonstrated similar results when fed diets supplemented with 250 and 500 mg/kg of thymol [20]. Unlike thymol, dietary yeast supplementation improved the growth performance and feed efficiency in the fish. Supporting these results, Zargham et al. [21] and Cid García et al. [22] reported growth promotion in rainbow trout when fed a diet supplemented with yeast. These benefits may be related to the components of the yeast cell wall, beta-glucans and mannan oligosaccharaide, which augment intestinal development and digestive enzyme activity [23].

Monitoring the ALT, AST and LDH levels in the blood circulation provides useful insight about the fish’s cellular health, as these enzymes are not functional in the blood circulation and originated from other tissues [24]. ALT is found at high concentrations in the fish hepatocytes, while LDH is found at high concentrations in skeletal muscles. The present results showed that thymol and yeast supplementation had no significant effects on hepatocytes and skeletal muscle cell health. On the other hand, AST is found in various cell types, and the present results suggest that the dietary yeast and thymol supplementation supported overall cell health in the fish body. It has been shown that dietary yeast supplementation at 0.1–2% induces no change in the blood ALT and AST levels in sea bream (*Sparus aurata*) [15], Galilee tilapia (*Sarotherodon galilaeus*) [25] and gibel carp (*Carassius auratus gibelio*) [26] but increases in these enzymes have been observed in the latter species when fed a diet containing 6% yeast.

Proper antioxidant defense protects biological materials against pro-oxidant agents. Immune responses trigger pro-oxidant formation and adequate antioxidant capacity contributes to maximum immune responses [27]. We found that dietary yeast and thymol supplementation improved antioxidant enzymes and suppressed lipid peroxidation. Supporting these results, dietary yeast administration significantly increased the activities of the antioxidant enzymes and decreased lipid peroxidation in Nile tilapia [28] and Ussuri catfish, *Pseudobagrus ussuriensis* [29]. Yeast cell wall components may have the main role in boosting the antioxidant function in fish, as demonstrated in previous studies [30,31]. Thymol is known as a powerful antioxidant compound, and studies on fish have shown that dietary thymol supplementation improves the activity of antioxidant enzymes and decreases lipid peroxidation, which generally led to improved fish stress and disease resistance (reviewed by Alagawany, Farag, Abdelnour and Elnesr [5]). Similar to the present study, dietary thymol supplementation significantly boosted the activities of SOD, CAT and GSH in various tissues of Nile tilapia. These changes were accompanied by reductions in lipid peroxidation and protein carbonylation following exposure to zinc nanoparticles [7]. Furthermore, pre-treating rainbow trout and snakehead with thymol-enriched diets improved the antioxidant capacity and enhanced the fish’s resistance to bacterial infections [6,8].

Improving the non-specific immunity suppresses the risk of pathogen localization in the fish body and disease outbreak, so one of the main goals of using functional feeds in aquaculture is boosting the fish’s innate immunity [32]. Lysozyme and complement proteins are two common players of the innate immune system; the first one kills Gram-positive bacteria [33] and the second one participates in foreign cell lysis, opsonization and inflammation [34]. Boosting lysozyme and complement activities in the blood circulation or mucosal surface can be beneficial in preventing disease outbreaks, as certain pathogens are capable of weakening these factors [17]. Fish resistant against experimental infection were found to have higher basal Ig concentrations, among other immunological factors, suggesting the possible participation of Ig in disease prevention at early stages [35,36]. The ALP in skin mucus has immunological functions such as the neutralization of pro-inflammatory compounds produced by pathogens [37]. We found improvements in plasma and skin mucus non-specific immune parameters, particularly in fish fed diets containing yeast. It has been reported that dietary yeast (or its extracted materials) administration improves humoral and skin mucus immune-related biochemical and transcriptional parameters in fish [38,39,40], which supports the present study. Likewise, dietary thymol supplementation resulted in improvements in humoral immunological parameters [6,8] but there are no data regarding its effects on skin mucus immunity.

*S. iniae*, *A. hydrophila* and *Y. ruckeri* are three common opportunistic pathogens threatening rainbow trout farms. The common route of pathogen transmission in the fish farms is the surrounding water [41]. Thus, well-functioning skin mucus immunity can mitigate the localization and entrance of these pathogens to the fish body [42]. After the entrance of pathogens to the fish body, humoral immune components can act as systemic tools for the elimination of the pathogens [32]. Therefore, an elevation of the bactericidal activities of the fish blood and mucus can be considered as indicators of the pathogen elimination efficiency. Studies have shown that dietary yeast [39,43] or thymol [6,8] supplementation improves fish skin mucus and humoral immune functions. Similarly, we found that yeast and thymol supplementation can improve the bactericidal activity in fish plasma; however, dietary thymol supplementation had no significant effects on the skin mucus bactericidal activity. On the other hand, both additives had greater effects on the plasma bactericidal activity than the skin mucus; thus, their systemic effect is higher than the superficial effect. Additionally, the increase in bactericidal activities may be associated with the high lysozyme and complement activities, although other factors such as antimicrobial peptides should not be neglected.

The major role of the fish hindgut is immunological, rather than being a digestive function. This part of the fish gut is in direct contact with the surrounding water and is threatened by waterborne pathogens [44]. Hence, a hindgut with boosted immunity is critical in preventing fish disease. Pro-inflammatory cytokines, such as *tnf-a* and *il-1b*, contribute to proper immune responses by triggering inflammation [45], and the present study showed that both thymol and yeast can up-regulate the expression levels of *tnf-a* and *il-1b* in the fish hindgut. There is no study regarding the effects of dietary thymol or yeast on *tnf-a* and *il-1b* expression levels in the fish hindgut under normal conditions. However, zebrafish (*Danio rerio*) larvae immersed in a solution containing thymol and carvacrol (Next Enhance 150; Novus International, Inc., Spain) exhibited an up-regulation in whole-body *il-1b* expression [46]. Additionally, dietary yeast hydrolysate supplementation resulted in significant up-regulation of the *tnf-a* and *il-1b* expression levels in the foregut and midgut of common carp, *Cyprinus carpio* [47]. The over-expression of pro-inflammatory cytokines results in drawbacks in the host, such as tissue damage and autoimmunity. Anti-inflammatory cytokines, for example TGF-b, are responsible for controlling inflammation [45]; hence, the up-regulation of *tgf-b* expression in thymol- or yeast-treated fish could be a controlling responses to up-regulation of the pro-inflammatory cytokines.

Defensins are anti-microbial peptides present in the mucosal surfaces, responsible for pathogen elimination and anti-inflammatory responses [48]. Studies have shown that the expression of *def-b* increases under microbial infection, both in the mucosal and systemic organs (reviewed by Das, Pradhan and Pillai [48]). The present study demonstrated that thymol and yeast supplementation can up-regulate the expression of this anti-microbial peptide in the fish hindgut, which may augment the hindgut health. Although there are no data regarding the effects of thymol or yeast on fish *def-b* expression, functional feeds have been found to up-regulate gut *def-b* expression. For example, dietary supplementation with certain amino acids [49,50,51], beta glucan [52] or organic acids [53] resulted in the up-regulation of *def-b* expression levels.

Proper folding of a protein guarantees its normal function, and molecular chaperones such as heat shock proteins are the main agent for this [54]. It has been found that functional feeds can increase *hsp-70* expression levels, accompanied by high stress tolerance by fish [53,55,56]. Similarly, the present study showed that thymol and yeast can improve *hsp-70* expression in the fish hindgut. There are few studies regarding the effects of dietary thymol and yeast on *hsp-70* expression in the fish gut. Zebrafish immersed in a solution containing thymol and carvacrol presented no change in whole-body *hsp70* expression [46]. However, thymol supplementation suppressed the up-regulation of *hsp70* expression in Nile tilapia under zinc nanoparticle toxicity, suggesting an attenuation in cell stress under unfavorable conditions [7]. Furthermore, dietary yeast supplementation induced no change in gut *hsp-70* expression but increased its expression levels in the liver and kidney of common carp [57].

## 5. Conclusions

In conclusion, dietary thymol supplementation does not promote growth in rainbow trout; however, it does enhance antioxidant and immunological parameters. In contrast, dietary yeast supplementation has been shown to improve growth performance, as well as antioxidant and immunological parameters. Notably, yeast exerts a more significant impact on immunological parameters than thymol. Furthermore, the inclusion of yeast in a diet containing 250 mg/kg of thymol results in improvements in gut-immune-related transcripts that are comparable to those observed in a diet containing 500 mg/kg of thymol alone.

## Figures and Tables

**Figure 1 animals-15-00302-f001:**
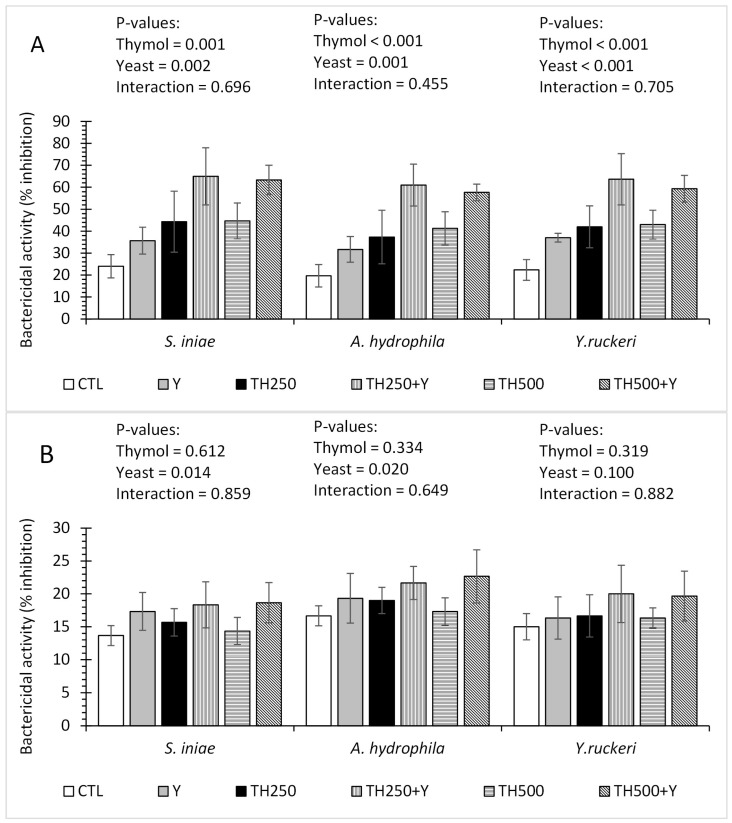
Bactericidal activity levels of plasma (**A**) and skin mucus (**B**) of the fish against *S. iniae*, *A. hydrophila* and *Y. ruckeri*. Data are shown as the mean ± SD of three replicates.

**Figure 2 animals-15-00302-f002:**
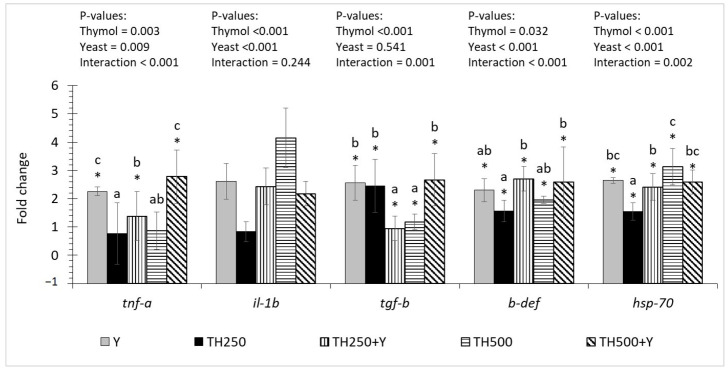
Relative expression levels of immune-related genes in the hind gut of the fish. Data are shown as mean ± SD of three replicates. Asterisks above the bars show significant up-regulations compared to the TH0-Y0 treatment. Different letters above the bars show significant differences.

**Table 1 animals-15-00302-t001:** Formulations and biochemical compositions of the diets.

Ingredients	Amount (g/kg)					
	CTL	Y	TH250	TH250+Y	TH500	TH500+Y
Fish meal ^1^	200	200	200	200	200	200
Wheat meal	200	200	200	200	200	200
Soybean meal ^2^	240	240	240	240	240	240
Poultry slaughterhouse by-product ^3^	270	270	270	270	270	270
Sunflower oil	40	40	40	40	40	40
Canola oil	40	40	40	40	40	40
Mineral premix ^4^	5	5	5	5	5	5
Vitamin premix ^5^	5	5	5	5	5	5
Biochemical composition						
Moisture (g/kg)	83.3	84.2	85.1	83.9	83.4	84.0
Crude protein (g/kg)	422	429	420	421	428	423
Crude fat (g/kg)	147	151	142	146	152	148
Crude ash (g/kg)	84.5	86.2	85.1	85.0	84.0	84.3
Crude fiber (g/kg)	19.2	19.3	18.4	18.0	18.9	19.3

^1^ Crude protein 67.9%; crude fat 10.2%; ^2^ crude protein 44.9%; crude fat 0.75%; ^3^ crude protein 54.6%; crude fat 17.3%; ^4^ Amineh Gostar Co. (Tehran, Iran). The premix provided the following amounts of vitamin to the diets (per kg): A: 1600 IU; D3: 500 IU; E: 20 mg; K: 24 mg; B3: 12 mg; B5: 40 mg; B2: 10 mg; B6: 5 mg; B1: 4 mg; H: 0.2 mg; B9: 2 mg; B12: 0.01 mg; C: 60 mg; Inositol: 50 mg; ^5^ Amineh Gostar Co. (Tehran, Iran). The premix provided the following amounts of minerals to the diets (per kg): Se: 0.15 mg; Fe: 2.5 mg; Co: 0.04 mg; Mn: 5 mg; Iodate: 0.05 mg; Cu: 0.5 mg; Zn: 6 mg; Choline: 150 mg.

**Table 2 animals-15-00302-t002:** Forward/reverse sequences of the primers.

Gene	Sequences		Amplicon	Efficiency	Annealing	Accession Number	Reference
*tnf-a*	F:	CAGGCTTCGTTTAGGGTCAAG	186	96.6	57 °C; 30 s	NM_001124357.1	[17]
	R:	AACTGCATTGTACCAGCCTTC					
*hsp-70*	F:	CGAGGATGGGATCTTTGAGGTG	131	96.9	57 °C; 30 s	XM_036940954.1	[17]
	R:	TCTGGCTGATGTCCTTCTTGTG					
*il-1b*	F:	GAAGTTGAGCAGGTCCTTGTC	146	97.1	57 °C; 30 s	AB118099.1	[17]
	R:	TCCACGAGCTGAAGAAAGAGA					
*b-def*	F:	GCGTTTCTAACCTGGCATGAT	145	95.2	57 °C; 30 s	NM_001195168.1	[17]
	R:	AACGGGATCCTCATAGCAGTT					
*tgf-b*	F:	TCTGAATGAGTGGCTGCAAG	75	96.6	57 °C; 30 s	X99303	[18]
	R:	GGTTTCCCACAATCACAAGG					
*gadph*	F:	GAGGGTCTGATGAGCACAGTTC	150	99.7	57 °C; 30 s	XM_021623341.2	[17]
	R:	GATGACCTTGCCGACAGCC					

**Table 3 animals-15-00302-t003:** Growth performance, survival and feed efficiency rates with different treatments.

	IW (g)	FW (g)	WG (%)	SGR (%/d)	FCR	Survival (%)
CTL	6.57 ± 0.29	26.0 ± 1.93	296 ± 14.2	2.29 ± 0.06	1.04 ± 0.05	100
Y	6.63 ± 0.12	27.5 ± 0.70	314 ± 12.3	2.37 ± 0.05	1.03 ± 0.04	100
TH250	6.70 ± 0.10	27.4 ± 0.80	310 ± 16.9	2.35 ± 0.07	1.06 ± 0.06	100
TH250+Y	6.53 ± 0.25	28.5 ± 0.64	337 ± 7.76	2.46 ± 0.03	0.95 ± 0.02	100
TH500	6.70 ± 0.26	29.7 ± 0.81	344 ± 7.90	2.48 ± 0.03	0.90 ± 0.02	100
TH500+Y	6.60 ± 0.10	29.6 ± 1.10	348 ± 22.9	2.50 ± 0.09	0.89 ± 0.06	100
*p*-value						
Thymol	0.578	0.104	0.265	0.260	0.421	-
Yeast	0.821	0.001	<0.001	<0.001	<0.001	-
Interaction	0.780	0.985	0.773	0.749	0.352	-

Data are shown as mean ± SD of three replicates. IW: initial fish weight; FW: final fish weight; WG: weight gain; SGR: specific growth rate; FCR: feed conversion ratio.

**Table 4 animals-15-00302-t004:** Plasma enzymatic activities in different treatments.

	ALT (U/L)	AST (U/L)	LDH (U/L)
CTL	20.3 ± 4.73	60.0 ± 7.55	203 ± 54.8
Y	15.3 ± 2.08	43.3 ± 6.65	149 ± 25.8
TH250	17.7 ± 1.53	39.3 ± 7.02	159 ± 22.6
TH250+Y	18.0 ± 2.65	46.3 ± 5.51	147 ± 14.5
TH500	15.0 ± 2.65	34.3 ± 5.86	141 ± 21.1
TH500+Y	16.0 ± 2.64	36.0 ± 7.94	138 ± 13.1
*p*-value			
Thymol	0.093	0.003	0.187
Yeast	0.309	0.019	0.056
Interaction	0.832	0.445	0.356

Data are shown as mean ± SD of three replicates. ALT: alanine aminotransferase; AST: aspartate aminotransferase; LDH: lactate dehydrogenase.

**Table 5 animals-15-00302-t005:** Hepatic antioxidant parameters of different treatments.

	MDA (nM/g·ww)	GSH (µM/g·ww)	TAC (µM/mg·pr)	SOD (U/mg·pr)	CAT (U/mg·pr)	GPx (U/mg·pr)
CTL	169 ± 17.0	2.21 ± 0.22	0.60 ± 0.06	16.1 ± 4.89	111 ± 9.54	135 ± 23.1
Y	118 ± 15.2	2.62 ± 0.37	0.72 ± 0.09	24.7 ± 4.04	145 ± 17.0	355 ± 19.3
TH250	113 ± 11.0	2.90 ± 0.26	0.79 ± 0.09	25.4 ± 5.84	147 ± 13.3	303 ± 12.2
TH250+Y	126 ± 8.88	2.72 ± 0.28	0.70 ± 0.06	22.0 ± 3.54	127 ± 10.7	205 ± 27.8
TH500	98.3 ± 7.51	3.17 ± 0.38	0.79 ± 0.05	27.3 ± 3.51	165 ± 12.7	346 ± 63.7
TH500+Y	98.7 ± 6.11	3.20 ± 0.38	0.83 ± 0.06	27.4 ± 5.14	164 ± 14.6	295 ± 29.5
*p*-value						
Thymol	<0.001	0.015	0.005	0.015	0.001	<0.001
Yeast	0.001	0.008	0.051	0.072	0.013	0.292
Interaction	0.121	0.758	0.784	0.825	0.965	0.106

Data are shown as mean ± SD of three replicates. MDA: malondialdehyde; GSH: reduced glutathione; TAC: total antioxidant capacity; SOD: superoxide dismutase; CAT: catalase; GPx: glutathione peroxidase.

**Table 6 animals-15-00302-t006:** Plasma and skin mucus immunological parameters of different treatments.

	Plasma			Skin Mucus		
	Lysozyme (U/mL)	ACH50 (%)	Total Ig (g/dL)	Lysozyme (U/mg·pr)	ALP (U/mg·pr)	Total Ig (%)
CTL	19.0 ± 3.00	21.3 ± 4.51	0.75 ± 0.09	28.7 ± 3.52	1.27 ± 0.17	35.7 ± 3.33
Y	23.3 ± 2.51	21.0 ± 3.00	0.81 ± 0.11	27.3 ± 6.02	1.58 ± 0.10	34.8 ± 7.65
TH250	22.3 ± 1.53	21.0 ± 2.00	0.87 ± 0.18	28.0 ± 3.51	1.58 ± 0.21	36.2 ± 5.22
TH250+Y	21.3 ± 3.05	25.7 ± 2.52	1.00 ± 0.12	33.7 ± 6.11	1.61 ± 0.13	37.8 ± 6.66
TH500	25.0 ± 2.00	31.3 ± 3.51	1.19 ± 0.19	35.3 ± 6.03	1.89 ± 0.15	35.0 ± 1.80
TH500+Y	26.7 ± 2.52	28.0 ± 3.61	1.23 ± 0.15	37.7 ± 7.37	1.96 ± 0.16	38.3 ± 3.25
*p*-value						
Thymol	0.019	0.395	0.126	0.854	0.006	0.821
Yeast	0.036	0.001	<0.001	0.015	0.001	0.573
Interaction	0.639	0.321	0.701	0.771	0.922	0.779

Data are shown as mean ± SD of three replicates. ACH50: alternative complement; Ig: immunoglobulin; ALP: alkaline phosphatase.

## Data Availability

The original contributions presented in the study are included in the article, further inquiries can be directed to the corresponding author.

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
