# Peer review of "Co-Supplementation of Diet with Saccharomyces cerevisiae and Thymol: Effects on Growth Performance, Antioxidant and Immunological Responses of Rainbow Trout, Oncorhynchus mykiss"

_animals, 2025, doi:10.3390/ani15030302_

Round 1

Reviewer 1 Report

Comments and Suggestions for Authors

The paper (animals-3437804) entitled "Co-supplementation of Diet with Saccharomyces cerevisiae and Thymol: Effects on Growth Performance, Antioxidant and Immunological Responses of Rainbow Trout, Oncorhynchus mykiss" mainly compared the individual and combined influence of dietary Saccharomyces cerevisiae and thymol on growth, antioxidant capacity, and immunoregulatory-related property in rainbow trout. The diet containing thymol had no significant growth-promoting effect, but it showed improved performance in immune responses in rainbow trout. The diet containing Saccharomyces cerevisiae exhibited superior growth and immune results in rainbow trout than those in the other diets.

The main content of this paper is meaningful to provide new insights into thymol and yeast as functional feed additives applied in farmed fish. Results from this study also could provide the reference for developing new, safe, effective feed additive with growth /health-improving properties in aquafeed.

However, the presentation of abstract and discussion in this manuscript are less clear and insufficient. The method part lacks of many critical information. Moreover, there are several mistakes in language, syntax, and format.

Therefore, the authors need to modify the corresponding parts of this paper to improve its quality.

Major comments:

1. Regarding the "Abstract" part, key information is missing, such as the initial size and total number of experimental fish, the primary results of growth performance. For example, Line 25. Thus, "Abstract" part should be clearly written and added more necessary information.

2. Regarding the "1.Introduction" section, the statements on research background were unclear and insufficient. For example, the brief background information on the experimental fish, rainbow trout (Oncorhynchus mykiss), should be provided using a concise summary.

Additionally, the texts at the 4th paragraph (Line 69-74) seemed to be more relevant to the main text of the 3rd paragraph (Line 60-68). It is suggested that the authors can merge these two parts of similar themes. Similar problem as Line 81-86, in which the descriptions with the relevant topic could be combined in the main text of Line 75-80.

Thus, the authors should rephrase the relevant statement of "1.Introduction" for a clear focus on background or significance in this study.

3. Regarding the section of "2.Materials and Methods", some methodological descriptions on key information are incomplete and unclear.

First, according to the text in Line 136-137, there was an acclimation period. So, how long did the acclimating period last?

Second, what about the aquatic environmental parameters (such as salinity, pH, dissolved oxygen, ammonia nitrogen, etc,) during the acclimating period and feeding trial? Are these parameters were same or not?

Third, in the original text of Line 137, no information on the initial body weight and body length was provided.

More importantly, the ethics statement is missing in this paper. It is recommended that the authors provide the ethical statement on animal experimentation in a separate part.

4. Regarding the "3.Results" part, the format of some tables (Table 3-6) is problematic. For example, the legend of Table 3 ("Data are shown as mean ± SD of three replicates") should be indicated below Table 3. Similar mistakes are present in the other table legend of the result part.

Moreover, it is suggested to provide a high-resolution image of Figure 1.

5. The main content of "4.Discussion" section is not well-organized and has many grammatical errors and statement issues, such as lengthy and confusing sentences, missing cited references, grammatical and stylistic errors, etc.

For example, in Line 361-362, the references are missing in the text of discussion part. The statement in Line 428-430 is overly verbose and readers tended to be more confused by this lengthy description. It would be preferable to split into two short sentences. Similar errors regarding over-long statements are present in the other part of discussion.

Moreover, the current text in the discussion part contain the limited comparisons on the interpretation of results. The discussion could be more in depth in terms of the similarities/differences of dietary thymol and yeast applied in aquatic amimals between the relevant researches, particularly diets containing different contents of thymol and/or yeast.

Thus, it is suggested to rephrase the main text in the discussion part for better emphasizing and clarifying your main findings.

Minor comments:

1. The current "Keywords" (Line 42-43) might not a good match to the main content of this manuscript. Please revise it by adding the correct terms and removing the redundant phrases. For example, "Saccharomyces cerevisia" and "thymole" should be included.

2. The authors need to check the current reference list of this manuscript. The current reference list is a bit chaotic, including wrong/missing volume and page numbers.

For example, in Reference 15, 36, the information on the volume and page number is incomplete. Additionally, the format of these two references is inconsistent compared with others. Please re-check and modify accordingly.

Other errors (highlighted in yellow) were marked in the PDF file.

So, this manuscript will be reconsidered after major revision.

Author Response

The paper (animals-3437804) entitled "Co-supplementation of Diet with Saccharomyces cerevisiae and Thymol: Effects on Growth Performance, Antioxidant and Immunological Responses of Rainbow Trout, Oncorhynchus mykiss" mainly compared the individual and combined influence of dietary Saccharomyces cerevisiae and thymol on growth, antioxidant capacity, and immunoregulatory-related property in rainbow trout. The diet containing thymol had no significant growth-promoting effect, but it showed improved performance in immune responses in rainbow trout. The diet containing Saccharomyces cerevisiae exhibited superior growth and immune results in rainbow trout than those in the other diets.

The main content of this paper is meaningful to provide new insights into thymol and yeast as functional feed additives applied in farmed fish. Results from this study also could provide the reference for developing new, safe, effective feed additive with growth /health-improving properties in aquafeed.

However, the presentation of abstract and discussion in this manuscript are less clear and insufficient. The method part lacks of many critical information. Moreover, there are several mistakes in language, syntax, and format.

Therefore, the authors need to modify the corresponding parts of this paper to improve its quality.

Major comments:

1-Regarding the "Abstract" part, key information is missing, such as the initial size and total number of experimental fish, the primary results of growth performance. For example, Line 25. Thus, "Abstract" part should be clearly written and added more necessary information.

Response: Thank you. We added this information to lines 26-27

2. Regarding the "1.Introduction" section, the statements on research background were unclear and insufficient. For example, the brief background information on the experimental fish, rainbow trout (Oncorhynchus mykiss), should be provided using a concise summary.

Response: Thank you. We tried to supplement this part of the manuscript. Please refer to lines 52-56

Additionally, the texts at the 4th paragraph (Line 69-74) seemed to be more relevant to the main text of the 3rd paragraph (Line 60-68). It is suggested that the authors can merge these two parts of similar themes. Similar problem as Line 81-86, in which the descriptions with the relevant topic could be combined in the main text of Line 75-80.

Response: Thank you. Following your comments, we revised this part of the introduction. Please refer to lines 72-74 and 85-87

Thus, the authors should rephrase the relevant statement of "1.Introduction" for a clear focus on background or significance in this study.

3. Regarding the section of "2.Materials and Methods", some methodological descriptions on key information are incomplete and unclear.

First, according to the text in Line 136-137, there was an acclimation period. So, how long did the acclimating period last?

Response: Thank you. It was 7 days. Please refer to lines 147-148

Second, what about the aquatic environmental parameters (such as salinity, pH, dissolved oxygen, ammonia nitrogen, etc,) during the acclimating period and feeding trial? Are these parameters were same or not?

Response: Thank you. Please check the lines 148-146 and 160-164, where we added this information.

Third, in the original text of Line 137, no information on the initial body weight and body length was provided.

Response: Thank you. We added the fish average weight in line 150. However, we did not measure fish length.

More importantly, the ethics statement is missing in this paper. It is recommended that the authors provide the ethical statement on animal experimentation in a separate part.

Response: Thank you. As per journal style, it should be at the end of the text. Please refer to lines 520-522

4. Regarding the "3.Results" part, the format of some tables (Table 3-6) is problematic. For example, the legend of Table 3 ("Data are shown as mean ± SD of three replicates") should be indicated below Table 3. Similar mistakes are present in the other table legend of the result part.

Response: Thank you. We revised the footnotes.

Moreover, it is suggested to provide a high-resolution image of Figure 1.

Response: Thank you. New images were provided.

5. The main content of "4.Discussion" section is not well-organized and has many grammatical errors and statement issues, such as lengthy and confusing sentences, missing cited references, grammatical and stylistic errors, etc.

Response: Thank you. We checked this part and fixed a number of writing issues.

For example, in Line 361-362, the references are missing in the text of discussion part.

Response: Please note that this sentence and the next one are jointed, so we inserted the reference after the second sentence.

The statement in Line 428-430 is overly verbose and readers tended to be more confused by this lengthy description. It would be preferable to split into two short sentences.

Response: Thank you. We considered your comment and revised it.

Similar errors regarding over-long statements are present in the other part of discussion.

Moreover, the current text in the discussion part contain the limited comparisons on the interpretation of results. The discussion could be more in depth in terms of the similarities/differences of dietary thymol and yeast applied in aquatic amimals between the relevant researches, particularly diets containing different contents of thymol and/or yeast. Thus, it is suggested to rephrase the main text in the discussion part for better emphasizing and clarifying your main findings.

Response: Thank you. We disagree the respectful reviewer, as we mentioned similar studies in all parts of the discussion. However we did not cite opposite studies in all case, as it is not useful for the readers and makes the discussion lengthy and boring.

Minor comments:

1. The current "Keywords" (Line 42-43) might not a good match to the main content of this manuscript. Please revise it by adding the correct terms and removing the redundant phrases. For example, "Saccharomyces cerevisia" and "thymole" should be included.

Response: thank you. We changed the keywords as you suggested.

2. The authors need to check the current reference list of this manuscript. The current reference list is a bit chaotic, including wrong/missing volume and page numbers. For example, in Reference 15, 36, the information on the volume and page number is incomplete. Additionally, the format of these two references is inconsistent compared with others. Please re-check and modify accordingly.

 Response: Thank you. It was related to a problem in my endnote software. I fixed it and the list is OK now.

Other errors (highlighted in yellow) were marked in the PDF file.

Response: Thank you. We applied all recommendations.

Reviewer 2 Report

Comments and Suggestions for Authors

Dear Authors,

As requested, I reviewed the manuscript “Co-supplementation of diet with Saccharomyces cerevisiae and thymol: effects on growth performance, antioxidant, and immunological responses of rainbow trout, Oncorhynchus mykiss” by Yousefi M, Adineh H, Mirghaed AT, Hosein SM. The work investigated dietary thymol, dietary yeast and dietary combined effects on growth and health of rainbow trout, with great focus on the diets with combined additives (this approach is poorly studied). The results showed improved effects exerted by yeast diets both alone and in combination with thymol.

The work is nice and quite interesting, and coherent with the aim of the journal. However, the paper shows few minor inaccuracies which are as follows:

-) In my opinion, the Introduction is poor of references. For example, I would suggest the authors to add references to lines 48-49, 50-51 and 75-76. Furthermore, when introduced the background of previous research performed on the investigation of thymol effects as dietary additive in fish species, poor literature has been added. The same was when described the example of studies dietary thymol on rainbow trout.

-) In line 47, the scientific name of rainbow trout should be added.

-) In Materials and Methods, the authors should better clarify the experimental design. It is not clear how fish have been divided in the tanks (number of fish per tank, number of replicates per group).

-) When describing the acclimation period, the duration should be added. Furthermore, when describing the rearing protocols, the parameter of water temperature, pH, and salinity, as well as the frequency of feeding (per day) should be added.

-) The authors should always add the unit of measure when a percentage is found.

-) In Paragraph “2.5 Immunological analysis”, it is not clear what the authors say with “as described before” (line 175).

-) In Paragraph “2.9 Hindgut gene expression”, the authors should say if tested primer couples were already used or if these have been designed. In first case, the authors should add the reference; in the second case, the authors should describe how primers have been designed.

-) In Table 2, in my opinion, the annealing temperature is a parameter to be added.

Thank you very much for your attention to my opinion.

Author Response

As requested, I reviewed the manuscript “Co-supplementation of diet with Saccharomyces cerevisiae and thymol: effects on growth performance, antioxidant, and immunological responses of rainbow trout, Oncorhynchus mykiss” by Yousefi M, Adineh H, Mirghaed AT, Hosein SM. The work investigated dietary thymol, dietary yeast and dietary combined effects on growth and health of rainbow trout, with great focus on the diets with combined additives (this approach is poorly studied). The results showed improved effects exerted by yeast diets both alone and in combination with thymol. The work is nice and quite interesting, and coherent with the aim of the journal. However, the paper shows few minor inaccuracies which are as follows:

-) In my opinion, the Introduction is poor of references. For example, I would suggest the authors to add references to lines 48-49, 50-51 and 75-76.

Response: Thank you. We added relevant references for the lines 48-49 and 50-51 (please refer to lines 59-61). The reference for the line 75-76 is similar to the 77-78, so we had added the reference at the end of these sentences (please refer to line 79-83)

Furthermore, when introduced the background of previous research performed on the investigation of thymol effects as dietary additive in fish species, poor literature has been added. The same was when described the example of studies dietary thymol on rainbow trout.

Response: Thank you. We disagree the respected reviewer. We first cited studies reported the effects of thymol on snakehead and Nile tilapia. We avoided to cite a study on grass carp, because the results of that study showed the fish were not kept under a good husbandry condition (high mortality and low growth). There is no study on other species, otherwise they used an herbal mixture containing thymol (as one of the ingredients). In the case of rainbow trout, there was only one study used pure thymol in rainbow trout diet, which we cited.

-) In line 47, the scientific name of rainbow trout should be added.

Response: Thank you. It was added.

-) In Materials and Methods, the authors should better clarify the experimental design. It is not clear how fish have been divided in the tanks (number of fish per tank, number of replicates per group).

Response: Thank you. Please refer to lines 152-154

-) When describing the acclimation period, the duration should be added. Furthermore, when describing the rearing protocols, the parameter of water temperature, pH, and salinity, as well as the frequency of feeding (per day) should be added.

Response: Thank you. We revised the text. Please refer to lines 147-150

-) The authors should always add the unit of measure when a percentage is found.

Response: Thank you. But I doubt I understand you clearly. Please explain more.

-) In Paragraph “2.5 Immunological analysis”, it is not clear what the authors say with “as described before” (line 175).

Response: we revised it. Please refer to line 200-201

-) In Paragraph “2.9 Hindgut gene expression”, the authors should say if tested primer couples were already used or if these have been designed. In first case, the authors should add the reference; in the second case, the authors should describe how primers have been designed.

Response: Thank you. Please note that these primers have been used in our previous works. We added the references.

-) In Table 2, in my opinion, the annealing temperature is a parameter to be added.

Response: Thank you. We added this to the table.

Round 2

Reviewer 1 Report

Comments and Suggestions for Authors

The revised manuscript (animals-3437804) under the title "Co-supplementation of Diet with Saccharomyces cerevisiae and Thymol: Effects on Growth Performance, Antioxidant and Immunological Responses of Rainbow Trout, Oncorhynchus mykiss" has been modified as suggested. The authors reply to the reviewer’s comments one by one (online system). Moreover, the corresponding explanation has been provided regarding the unchanged part in the revised manuscript.

The current revision is suitable for publication in your journal, though it still contains some minor errors. For example, in Line 22, replace "Yeast" with "yeast". The journal style of Page 1 (title page) should be revised according to the journal template of Animals.